# Transcriptome Analysis Reveals the Involvement of Mitophagy and Peroxisome in the Resistance to QoIs in *Corynespora cassiicola*

**DOI:** 10.3390/microorganisms11122849

**Published:** 2023-11-23

**Authors:** Bingxue Sun, Rongjia Zhou, Guangxue Zhu, Xuewen Xie, Ali Chai, Lei Li, Tengfei Fan, Baoju Li, Yanxia Shi

**Affiliations:** State Key Laboratory of Vegetable Biobreeding, Institute of Vegetables and Flowers, Chinese Academy of Agricultural Sciences, Beijing 100081, China; sunbingxuechina@163.com (B.S.); j1824967583@163.com (R.Z.);

**Keywords:** mitophagy, peroxisome, reactive oxygen species

## Abstract

Quinone outside inhibitor fungicides (QoIs) are crucial fungicides for controlling plant diseases, but resistance, mainly caused by G143A, has been widely reported with the high and widespread use of QoIs. However, two phenotypes of *Corynespora casiicola* (RI and RII) with the same G143A showed significantly different resistance to QoIs in our previous study, which did not match the reported mechanisms. Therefore, transcriptome analysis of RI and RII strains after trifloxystrobin treatment was used to explore the new resistance mechanism in this study. The results show that 332 differentially expressed genes (DEGs) were significantly up-regulated and 448 DEGs were significantly down-regulated. The results of GO and KEGG enrichment showed that DEGs were most enriched in ribosomes, while also having enrichment in peroxide, endocytosis, the lysosome, autophagy, and mitophagy. In particular, mitophagy and peroxisome have been reported in medicine as the main mechanisms of reactive oxygen species (ROS) scavenging, while the lysosome and endocytosis are an important organelle and physiological process, respectively, that assist mitophagy. The oxidative stress experiments showed that the oxidative stress resistance of the RII strains was significantly higher than that of the RI strains: specifically, it was more than 1.8-fold higher at a concentration of 0.12% H${}_{2}$O${}_{2}$. This indicates that there is indeed a significant difference in the scavenging capacity of ROS between the two phenotypic strains. Therefore, we suggest that QoIs’ action caused a high production of ROS, and that scavenging mechanisms such as mitophagy and peroxisomes functioned in RII strains to prevent oxidative stress, whereas RI strains were less capable of resisting oxidative stress, resulting in different resistance to QoIs. In this study, it was first revealed that mitophagy and peroxisome mechanisms available for ROS scavenging are involved in the resistance of pathogens to fungicides.

## 1. Introduction

Fungi are important plant pathogens—with more than 8000 species known to cause plant diseases—that account for about 70–80% of all plant diseases [1,2,3,4]. Fungal diseases can cause yield and quality loss, while some fungi can also secrete toxins that pose a significant threat to the safety of agricultural products [5,6,7,8]. Measures against fungal diseases include disease-tolerant varieties, agricultural control, biological control, and chemical control, of which, the most effective and rapid method is the use of fungicides [9,10]. There are now 55 classes of fungicides reported in FRAC, such as methyl-benzimidazole carbamates (MBCs), demethylation inhibitors (DMIs), dicarboximides (DCFs), and succinate dehydrogenase inhibitors (SDHIs), of which, the quinone outside inhibitor fungicides (QoIs) are extremely important for a variety of disease control (http://www.chinapesticide.org.cn/, accessed on 9 February 2023).

QoIs are fungicides that act on the mitochondrial electron transport chain to inhibit electron transport and ATP production [11,12]. Due to widespread and extensive use, resistance to QoIs has been found in 35 kinds of pathogens, such as *Mycosphaerella graminicola*, *Zymoseptoria tritici*, *Corynespora cassiicola*, *Podosphaera xanthii*, *Cercospora beticola*, *Magnaporthe oryzae*, *Alternaria alternata*, *Venturia inaequalis*, and *Plasmopara viticola* [13,14,15,16,17,18,19,20,21]. The frequency of resistance is generally above 50% and tends to increase—especially, more than 95% for *C. cassiicola*—to QoIs in China [22]. The mechanisms of fungal resistance to QoIs have been reported to include: (i) Mutations in the *Cytb* genes, where mutations in amino acid sites lead to decreased affinity and weakened binding of target proteins to the agent. The mutations are mainly G143A, F129L, and G137R, of which G143A has been found in more than 20 pathogens, such as *Pseudoperonospora cubensis*, *Botrytis cinerea*, and *C. casiicola* (Fungicide Resistance Action Committee (FRAC) www.frac.info (accessed on 21 September 2023)). (ii) Alternative respiration, which is mediated by the induction of alternative, cyanide-resistant respiration, which is maintained by alternative oxidase (AOX) [23,24,25]. For example, *M. oryzae* induces AOX respiration and resulting resistance when metominostrobin blocks the cytochrome electron transport pathway [26]. (iii) Overexpression of transporters, which prevents fungicides in pathogens from reaching lethal concentrations to reduce bactericidal efficacy [27,28,29]. For example, overexpression of the transporter protein AtrB resulted in multidrug resistance to fludioxonil, cyprodinil, and tolnaftate in *B. cinerea* [30].

In medicine, mechanisms such as target gene mutation and overexpression of transporters have also been reported. However, there is a common resistance mechanism in medicine that has not been reported more in fungicides. This extremely important mechanism is that cancer cells can regulate drug resistance by modulating physiological processes and organelles such as mitophagy and peroxisome for which ROS is a major inducer [31,32,33].

ROS production in excess causes oxidative stress and produces irreversible reactions such as lipid peroxidation, protein oxidation, and DNA damage, which induce a series of intracellular immune responses, resulting in reduced levels of the agent and leading to reduced inhibitory effects [34,35,36]. The main ROS scavenging mechanisms that have been reported are mitophagy and peroxisome-mediated reactive oxygen species regulatory pathways [31,32,33]. Mitophagy, a process that relies on the lysosome to degrade damaged mitochondria through selective autophagy, plays an important role in cell differentiation, inflammation, and apoptosis [37,38]. Especially in drug resistance, the quality and quantity of mitochondria can be regulated through mitophagy, which protects the survival of cells to some extent, creating resistance to both cancer and fungicides. In medicine, resistance caused by increased or inhibited mitophagy has been reported in hepatocellular carcinoma cells, colon cancer cells, myeloma cells, breast cancer cells, and due to chemotherapy [39,40,41,42]. It has been found that activation of mitophagy doubles the resistance of lung cancer cells to cisplatin [43]. And inhibition of mitophagy increased the sensitivity of human cervical cancer HeLa cells to cisplatin by about 80% [44]. Even for multidrug-resistant cancer cells, the sensitivity to B5G1 was doubled after the inhibition of their mitophagy [32].

Peroxisomes play an important role in the cell, especially in the scavenging of ROS. The peroxisome contains several enzymes, the most important of which are catalase (CAT), superoxide dismutase (SOD), and glutathione (GSH). Among them, SOD converts superoxide into H${}_{2}$O${}_{2}$ and O${}_{2}$ by accelerating its decomposition, and CAT and GSH convert H${}_{2}$O${}_{2}$ into harmless water and O${}_{2}$ by speeding up the breakdown of H${}_{2}$O${}_{2}$, which degrade ROS into harmless substances, maintain intracellular redox balance, and protect cells from oxidative damage [33,45,46,47]. Studies have reported that SOD is associated with resistance production and that inhibition of SOD expression and activity reduces the resistance of colorectal cancer cells to anticancer drugs [48]. GSH has also been reported to be associated with the development of resistance, e.g., increased GSH in cells increases resistance to cisplatin [49].

In our previous study, we identified two phenotypes of *C. cassiicola* (RI and RII) with G143A mutations, which exhibited significantly varying resistance to QoIs. However, it was found to be impossible to explain the emergence of this phenomenon by common fungicide resistance mechanisms (target gene point mutation, alternative respiration, or overexpression of transporters). Therefore, this study aimed to investigate the resistance mechanism and related metabolic pathways using transcriptome analysis, aiming to provide valuable insights into the control of fungal infections.

## 2. Materials and Methods

### 2.1. Fungal Isolates

Two phenotype strains of *C. cassiicola* (RI and RII) both carried the G143A mutation but differed in resistance to QoIs. Six RI strains (Cc27, Cc53, Cc98, Cc226, Cc260, and Cc241) and six RII strains (Cc5, Cc13, Cc66, Cc71, Cc120, and Cc177) were randomly selected for subsequent experiments. These isolates were preserved by the innovative team for vegetable disease control at the Chinese Academy of Agricultural Sciences.

### 2.2. Sensitivity Test to Fungicides

To determine the susceptibility of RI and RII strains to QoIs, four fungicides were selected: trifloxystrobin (Tri), kresoxim-methyl (Kre), azoxystrobin (Azo), and pyraclostrobin (Pyr), which target the Qo site of the cytochrome bc1 complex. Additionally, seven other fungicides were selected to determine susceptibility: cyazofamid (Cya, targeted to the Qi site of the cytochrome bc1 complex), penthiopyrad (Pen, which targets the ubiquinone binding site), terbinafine (Ter, which inhibits lanosterol 14α-demethylase), fludioxonil (Flu, which is involved in the high-osmolarity glycerol (HOG) stress response signal transduction pathway), tolnaftate (Tol, which inhibits ergosterol production), difenoconazole (Dif, which inhibits fungal lanosterol-14α-demethylase activity and blocks ergosterol biosynthesis), and carbendazim (Car, which hinders microtubule assembly and disrupts spindle formation) [50,51,52,53,54,55]. Among them, Car was dissolved with 0.2 M HCL and the other fungicides were dissolved with DMSO. The concentrations of active ingredients used were as follows: Kre and Cya (0, 0.1, 1, 10, 100, and 300 μg/mL); Azo and Pyr (0, 0.1, 1, 10, 80, and 240 μg/mL); Ter (0, 0.001, 0.01, 0.1, 1, and 5 μg/mL); Flu (0, 0.005, 0.01, 0.05, 0.1, and 0.5 μg/mL); Dif (0, 0.05, 0.1, 0.5, 3, and 9 μg/mL); Car (0, 10, 50, 200, and 400 μg/mL); and Pen (0, 0.1, 0.5, 1, 3, and 10 μg/mL).

The sensitivity of *C. cassiicola* to Tol, Flu, Ter, Dif, and Car was determined by PDA (200 g/L potato, 20 g/L dextrose, 20 g/L agar) medium. And the sensitivity of *C. cassiicola* to QoIs was determined by YBA agar (10 g/L yeast extract, 10 g/L peptone, 20 g/L sodium acetate, 15 g/L agar) medium. SHAM (50 μμg/mL) was added to the YBA agar medium for sensitivity measurements to QoIs. Mycelial plugs obtained from 5-day-old colony margins were placed on medium plates (60 mm diameter) containing different fungicide concentrations. After the plates were incubated in the dark at 28 °C for 5 days, the diameters of the colonies were measured. Each concentration had three replicates for each isolate, and the experiment was repeated twice. The EC${}_{50}$ values were calculated by probit regression of the computer software SPSS and Duncan multiple range tests were analyzed by using one-way analysis of variance (ANOVA); the significance level of the test was considered as *p*-value < 0.05.

### 2.3. Total RNA Extraction and RNA-Seq

The RI and RII strains were cultured on PDA at 28 °C for 5 days. Mycelium was extracted and shaken using YEPD (10 g/L yeast extract, 20 g/L peptone, 20 g/L dextrose) at 28 °C 180 rmp for 2 days and continued for 12 h with the addition of Tri. Mycelium obtained by filtration using triple filter paper was ground in liquid nitrogen, and total RNA was extracted with TRIzol${}^{@}$ reagent (Invitrogen, Waltham, CA, USA) using the manufacturer’s instructions. The mRNA was enriched by oligo (dT) magnetic beads from the total RNA. Subsequently, one strand of cDNA was synthesized by reverse transcription with six-base random hexamers using mRNA as the template. Two-strand cDNA was synthesized by adding buffer, dNTPs, and DNA polymerase I. Then, double-strand cDNA was purified by using AMPure XP beads. Modification of the purified double-stranded cDNA (Beijing Allwegene Technology Co., Ltd, Beijing, China). Fragment size selection of the double-stranded cDNA was performed by AMPure XP beads (Target Technology (Beijing) Co., Ltd., Beijing, China), and PCR amplification was performed to construct cDNA libraries. Finally, the library quality was assessed using an Agilent bioanalyzer 2100 system (Agilent Technologies Co. Ltd., Beijing, China); the samples were high-throughput sequenced after qualification.

### 2.4. Identification of DEGs and Assessment of GO and KEGG Enrichment

The HTSeq (v 0.5.4) software was used to analyze the gene expression of each sample, and an FPKM value of 0.1 or 1 was used as the threshold value to determine whether a gene was expressed or not. The threshold for screening differential genes using DEseq for comparison was: |log2 (Fold Change)| > 0 and padj < 0.05. For differential genes, if the |log2 (Fold Change)| > 0, the differential gene was considered to be up-regulated. If |log2 (Fold Change)| < 0, the differential gene was considered to be down-regulated. GO functional annotation and KEGG enrichment analysis were performed by comparing sequences with public databases. The functions of DEGs were described using the GO database, and enrichment analysis was performed using GOseq software. Pathway analysis of DEGs was performed using the KEGG database to identify the most important biochemical metabolic pathways and signal transduction pathways involved in DEGs. The *p*-value was corrected using Benjamini and Hochberg FDR, and smaller, corrected *p*-values represented more significance; a *p*-value less than 0.05 was defined as a pathway significantly enriched for the target gene.

### 2.5. Quantitative RT-PCR (qPCR) Analysis

To verify the accuracy of the transcriptomics data, six genes were randomly selected for qPCR analysis. Gene-specific primers were designed by Primer Premier 5.0, with elongation factor 1 alpha (EF-1 alpha) as an internal reference gene. Each sample was repeated three times. Further, qPCR was performed in a system containing 2 × AceQ Universal SYBR qPCR Master Mix 10.0 μL, 0.4 μL of each primer (10 μM), 1 μL of template DNA, and ddH${}_{2}$O to supplement to 20 μL. The qPCR reaction conditions were 95 °C for 5 min, 40 × (95 °C for 10 s, 60 °C for 30 s), 95 °C for 15 s, 60 °C for 1 min, and 95 °C for 15 s. Three replicates of each sample were performed, and the relative expression levels were calculated and analyzed using the $2^{-\Delta CT}$ method, with EF-1 alpha as the internal reference gene correction.Finally, Duncan multiple range test of one-way ANOVA was used to compare significant differences between means; the significance level of the test was considered as *p*-value < 0.05.

### 2.6. Sensitivity to Oxidative Stress

In order to clarify the relationship between oxidative stress and the difference in resistance to QoIs in RI and RII type strains, the oxidative stress assay was performed by the previous method (Sun et al., 2022) [56]. Sensitivity was determined using H${}_{2}$O${}_{2}$ (0, 0.12%, 0.24%, 0.36%) and 5 mM paraquat, while PDA plates without any agents served as controls. The diameter of the colonies was measured following 7 days of incubation. The mycelial radial growth inhibition (PIMG) was calculated as PIMG = [(C − N/(C − 5)] × 100, where C is the diameter of the untreated control colony (mm) and N is the diameter of the agent treatment (mm).

## 3. Results

### 3.1. The Sensitivity of RI and RII Strains to Fungicides

To assess the sensitivity to QoIs and other fungicides, the resistances of RI and RII strains to 11 fungicides were determined. The results showed that RII strains were significantly more resistant to QoIs than RI strains. The average EC${}_{50}$ values of RII strains were 909.44, 8565.47, 54.69, and 5.55 μg/mL for 4 QoIs (Tri, Kre, Azo, and Pyr), while the average EC${}_{50}$ values of RI strains were 14.49, 1.21, 11.15, and 0.85 μg/mL for 4 QoIs, respectively. Among them, the largest difference in the resistance was observed in Kre, followed by Tri, with the average EC${}_{50}$ of RII strains being 7102.38 and 62.78 times higher, respectively, than that of RI strains. And the resistance of RII strains to Azo and Pyr was 4.91 and 6.51 times higher, respectively, than that of RI strains. However, there was no significant difference in resistance between RI and RII strains to the other seven fungicides (Cya, Tol, Ter, Flu, Dif, Car, and Pen), with the average EC${}_{50}$ values being 8529.00, 1.46, 0.37, 0.12, 2.01, 194.90, and 8.88 μg/mL, respectively, for RII strains and 389.50, 23.69, 0.67, 0.12, 2.88, 341.72, and 0.80 μg/mL, respectively, for RI strains (Figure 1). Note that due to the extreme resistance to Cya, the EC${}_{50}$ data are only for reference (formulaic extrapolation based on inhibition at low concentrations).

### 3.2. Sequencing Data Quality Assessment and Sequence Comparison

In this study, the two phenotypes of *C. cassiicola* strains treated with Tri were sequenced by RNA-seq. The number of bases (clean base) of each sample was higher than 7.5 G, while the correlations between the three biological replicates were all greater than 0.93 for each sample. The quality of Q20 was higher than 97% (97.69–97.99%), Q30 was higher than 93% (93.88–94.91%), and the GC content was higher than 56% (56.64–57.36%) (Table 1). The above results show the quality of sequencing and the amount of data qualified to perform biological analysis.

### 3.3. Identification and Analysis of Differentially Expressed Genes (DEGs)

The Venn graph intuitively shows the overlap of DEGs by comparing the four strains (Cc71-Tri (RII) vs. Cc241-Tri (RI), Cc71-Tri (RII) vs. Cc260-Tri (RI), Cc66-Tri (RII) vs. Cc241-Tri (RI), and Cc66-Tri (RII) vs. Cc260-Tri (RI)). There were 1152 common DEGs found in the four groups simultaneously, with 448 DEGs down-regulated and 332 DEGs up-regulated (Figure 2).

In addition, the volcanic plots were analyzed and screened for DEGs by comparing the four strains two-by-two. The results indicated that there were 4614 DEGs between Cc66-Tri (RII) vs. Cc241-Tri (RI), with 2299 DEGs up-regulated and 231 down-regulated. A total of 5010 DEGs were screened between Cc66-Tri (RII) vs. Cc260-Tri (RI), including 2480 up-regulated genes and 2530 down-regulated genes. There were 4125 DEGs between Cc71-Tri (RII) vs. Cc241-Tri (RI), including 2033 up-regulated and 2092 down-regulated genes. Between Cc71-Tri (RII) and Cc260-Tri (RI), 4144 DEGs were identified, including 1906 up-regulated and 2238 down-regulated genes (Figure 3).

### 3.4. KEGG Enrichment Analysis

In order to further understand the functions of these genes, KEGG enrichment was applied to identify pathways of DEGs, which were also combined and classified. The KEGG enrichment analyses results of RI and RII strains revealed that the DEGs were enriched in a total of 95 KEGG pathways. Among these pathways, ribosomes had the highest enrichment factor and the metabolic pathway had the highest number of DEGs, with a total of 115 genes being enriched (Table A1 and Figure 4). In addition, in the enrichment of down-regulated and up-regulated genes, pathways such as autophagy, mitophagy, and MAPK signaling pathways were found to appear in the top 20 of the enrichment list (Figure 5). This was further analyzed for comparison between RI and RII strains, with glutamate, glycine, and cysteine metabolism (22 DEGs); peroxisome (3 DEGs); autophagy (4 DEGs); mitophagy (4 DEGs); endocytosis (2 DEGs); and glutathione metabolism (3 DEGs) identified (Figure 4).

Similarly, two-by-two comparative KEGG analyses were performed for each strain in which the highest enrichment factor was all in ribosomes and the highest number of enriched DEGs were in metabolic pathways. The peroxisomes were found in Cc66-Tri (RII) vs. Cc241-Tri (RI), Cc66-Tri (RII) vs. Cc260-Tri (RI), Cc71-Tri (RII) vs. Cc241-Tri (RI), and Cc71-Tri (RII) vs. Cc260-Tri (RI) to be enriched in 28, 28, 15, and 35 DEGs, respectively. Autophagy was enriched in 38, 42, 26, and 24 DEGs, respectively. Mitophagy was enriched in 12, 18, 14, and 9 DEGs, respectively. The endocytosis was enriched in 18, 18, 25, and 28 DEGs, respectively. And glutamate, glycine, and cysteine metabolism were enriched in 80, 76, 70, and 59 DEGs, respectively (Figure 6).

### 3.5. GO Functional Annotation

GO functional enrichment analysis was performed to analyze the different functions of DEGs in RI vs. RII groups, for which 1152 DEGs were enriched into 1968 GO terms. The top 20 enriched terms were found to be associated with DNA integration, ADP-binding, and metabolism and biosynthesis of amide, protein, peptide, and compounds (Table A2 and Figure 7). But these genes have not been reported to be closely related to resistance production, so similar terms were categorized with reference to KEGG (Figure 7). The results revealed that peroxisomes, endocytosis, and the lysosome were all enriched for DEGs. Among them, peroxisomes had the most DEGs, with 16 terms.

In addition to this, a two-by-two comparison of the strains was also performed (Figure 8). The results indicated that there were 107,996 DEGs between Cc66-Tri (RII) vs. Cc260-Tri (RI), with 49,292 up-regulated DEGs and 558,704 down-regulated DEGs. There were 101 DEGs for peroxide, endocytosis, the lysosome, and autophagy (including three CAT-related genes, one SOD-related gene, and some genes like acyl-CoA oxidase); 4 associated with some hypothetical-protein-related genes; 3 related to the OPT superfamily oligopeptide transporter, amino acid permease, and kinase-like protein; and 16 including genes such as the mitotic spindle checkpoint protein MAD2 and kinase-like protein. There were 102,137 DEGs between Cc66-Tri (RII) vs. Cc241-Tri (RI), including 2033 up-regulated and 49,887 down-regulated genes. And for Cc66-Tri (RII) vs. Cc241-Tri (RI), GO enrichment identified 98 DEGs of peroxide (including 2 SOD-related genes and 3 CAT-related genes), 5 DEGs of endocytosis (including a ClpP/crotonase-related gene), 4 lysosomal DEGs (including 2 OPT-superfamily-oligopeptide-transporter-related genes), and 24 DEGs of autophagy (including genes for the mitotic spindle checkpoint protein MAD2 and autophagy). There were 91,586 DEGs for Cc71-Tri (RII) and Cc241-Tri (RI), of which 49,142 were up-regulated and 42,444 were down-regulated. In addition, there were 59 DEGS for peroxide, endocytosis, the lysosome, and autophagy (including 3 CAT-related genes, 1 SOD-related gene, and 4 genes for peroxidase); 5 DEGS for some hypothetical-protein-related genes; 4 DEGS for genes related to cytochrome P450, OPT-superfamily-oligopeptide-transporter-related genes, and amino acid permease; and 9 DEGS for some hypothetical-protein-related genes. There were 87,649 DEGs for Cc71-Tri (RII) vs. Cc260-Tri (RI), with 47,296 up-regulated genes and 40,353 down-regulated genes. There were 98 DEGs for peroxide, endocytosis, the lysosome, and autophagy (including 1 SOD-related gene, 4 CAT-related genes, and some genes for oxidoreductase and peroxidase), 4 were included in the ClpP/crotonase-related gene, 1 gene for the OPT superfamily oligopeptide transporter, and 9 genes for autophagy and DUF1649-domain-containing protein.

### 3.6. Real-Time Quantitative PCR (qRT-PCR) Validation of Transcriptomic Dates

The accuracy of the transcriptome data was verified by qRT-PCR, for which six genes were randomly selected for validation: A1623, A7233, A0831, A5647, A2341, and A0004; these were found to be associated with resistance development in our previous studies and were being simultaneously studied. The expression of A1623 and A7233 was significantly higher in the RII strains than in the RI strains (Figure 9). In contrast, the expression of A2341, A0004, and A5647 in the RI strains was significantly higher than that of the RII strains, with more than a 10-fold difference in expression. This is consistent with the transcriptome data and validates the reliability of the results of this study. However, it is important to note that this result can only be used as a reference and does not fully represent the changes to the protein or resistance mechanism, which still need to be verified by subsequent proteomics or other molecular tests.

### 3.7. Oxidative Stress

The inhibition rate of RII strains was lower than that of RI strains under oxidative stress. The inhibition rates of RI strains were found to be 1.88, 1.31, and 1.19 times higher than those of RII strains at H${}_{2}$O${}_{2}$ concentrations of 0.12%, 0.24%, and 0.36%, respectively. Particularly, at 0.12% and 0.24% concentrations, the two phenotypic strains exhibited significantly different inhibition rates. The inhibition rate of RI strains was also significantly lower than those of the RII strains after treatment with paraquat (Figure 10). This indicated that the RI strains were significantly more sensitive to oxidative stress than the RII strains, resulting in different resistance to QoIs.

## 4. Discussion

With the use of QoIs, widespread resistance to them has been developed in *C. cassiicola*. Many studies have reported that resistance to QoIs is mainly caused by mutations [57,58,59]. Interestingly, we found significant differences in resistance to QoIs between two phenotypes of strains of *C. cassiicola* that each had G143A mutations. Transcriptome data analysis was performed to investigate the mechanism by which this phenomenon arose, which showed that the DEGs of RI and RII strains were mainly enriched in ribosomes but were also found to be enriched in mitophagy, peroxisome, lysosomes, and endocytosis. Oxidative stress experiments demonstrated that resistance to oxidative stress differed significantly between RI and RII strains, revealing the involvement of ROS in the development of resistance in the fungus.

Mitophagy is an important autophagic process in cells and is primarily responsible for clearing aged or damaged mitochondria to maintain the quality of mitochondria within the cell [60,61]. In drug resistance, mitophagy maintains or even enhances normal physiological functions to a certain extent, leading to resistance to cancer and fungicides [62,63,64]. In medicine, reports have focused on various cancer drugs and chemotherapy [39,65,66]. The main agents for which resistance is generated by mitophagy are sorafenib and cisplatin, which usually leads to about doubling of resistance [43]. For example, the knockdown of mitogenic genes increased the sensitivity of hepatocellular carcinoma cells to sorafenib by 65–100% [67]. In fungicides, there are few studies on the development of resistance by mitophagy, which has only been reported to be involved in azole fungicide resistance [68]. In our study, some genes differed significantly between RI and RII strains. The significant up-regulation of MDM34, known as a mitophagy-specific gene that relies on ubiquitination for efficient mitophagy, indicates a significantly higher level of mitophagy in the RII strains compared to the RI strains [69]. ATG27, which is an autophagy-related protein involved in vesicle formation, has been reported to be associated with antifungal agent resistance in *Candida albicans* [70,71]. Its up-regulation suggests that autophagy was up-regulated and enhanced resistance to QoIs in RII strains. HOG1 has been reported to be involved in the MAPK pathway, encoding mitogen-activated protein kinase, which regulates biological processes such as apoptosis and stress adaptation [72,73]. Its up-regulation indicates enhanced mitophagy in RII strains, which leads to increased resistance. MSS4 and ElF2α, which prevent programmed cell death, were also found to be significantly up-regulated, indicating that cell survival was maintained, verifying that mitophagy plays a role [74,75]. Therefore, the enrichment of these validates the occurrence of oxidative stress response and mitophagy.

The peroxisome responds to oxidative stress caused by ROS mainly by reducing intermediates and peroxides through SOD, CAT, and ascorbate-glutathione (ASC-GSH), which reduce irreversible reactions such as DNA damage and lipid peroxidation as a result of ROS overload [76,77,78]. Peroxisomes have been reported to be associated with the development of drug resistance in both cancer cells and fungicides [79]. In fungicides, the functional characterization of a glutathione peroxidase homolog of inactivation increased resistance to vinclozolin or fludioxonil for *A. alternata* [33]. CAT has similarly been reported to be intimately associated with resistance development. The inactivation of CAT makes a wide range of pathogens and *Pseudomonas aeruginosa* highly sensitive to the disinfectant solution H${}_{2}$O${}_{2}$ [80]. The knockout of *SsCat2* (encoding CAT) reduced the sensitivity level of the strain to QoIs by 50% [81]. The transcriptome of this study showed that two major CAT genes were significantly up-regulated in RII strains. And significant up-regulation of the EPHX2 gene was found, which was reported to reduce ROS levels and apoptosis rate [82]. This suggests that CAT in the peroxisome plays an important role in the development of resistance to QoIs in fungi.

Lysosomes carry many hydrolytic enzymes and proteins, which are key organelles for the degradation of heterophonic and autophagic contents, etc. [83,84,85]. Therefore, the production and activity of lysosomes plays an important role in promoting mitophagy. In addition, lysosomes have also been reported to be involved in the development of drug resistance. It has been shown that lysosomes contribute to the resistance to hydrophobic weak-base chemotherapeutic drugs mainly through the mechanism of lysosomal sequestration. Lysosomal sequestration is a phenomenon based on cation trapping. When encountering the acidic environment within the lysosome, the drugs become protonated and can no longer cross the lipid membrane. This leads to an apparent accumulation of drugs in the lysosomes, but the target sites cannot be found; thus, their ability to exert cytotoxic effects is hindered [86,87,88,89]. Currently, many agents have been reported to have developed resistance due to lysosomal sequestration, such as doxorubicin, mitoxantrone, fluoxetine, and vincristine [90,91,92,93]. Moreover, it has been found that an increase in the number of lysosomes per cell may be a marker of resistance to hydrophobic weak-base drugs, which tend to accumulate significantly in lysosomes [89]. The GO enrichment in this study revealed significant differences in the expression of genes important for lysosomal translocation in the two strains. Therefore, the enrichment of DEGs for lysosomes confirms the occurrence of mitophagy and that lysosomal sequestration may have exacerbated the resistance of the *C. cassiicola* to QoIs.

Endocytosis regulates many processes of cell signaling by controlling the number of functional receptors that are used specifically to take up extracellular proteins or other compounds on the cell surface [94,95,96]. It can deliver damaged substances into lysosomes, thereby promoting mitophagy. It has been shown that resistant cancer cells have fewer receptors for endocytosis and more rapid degradation than sensitive cancer cells. This suggests that endocytosis is reduced in resistant cells, which could also be based on a series of results triggered by lysosomal acidification [97]. In addition, endocytosis was found to cause resistance to Trastuzumab Emtansine (T-DM1) [98]. In this study, the PIP5K gene was found to be significantly up-regulated and was reported to regulate phosphorylation and participate in cytokinesis, endocytosis, and so on [99]. In addition, significant up-regulation of VPS22 was found, which is reported to be involved in sorting endocytosed ubiquitinated receptors to lysosomes for degradation and efficient termination of signaling [100]. Thus, the enrichment of endocytosis reinforces the occurrence of mitochondrial autophagy and may be a novel drug resistance mechanism.

Metabolism-related genes are widely involved in cellular energy metabolism, substance synthesis, catabolism, and transport, which can affect the cell’s ability to adapt to the external environment. It has been demonstrated that polysaccharide metabolites, lipid metabolites, and certain functional genes, such as the R gene, can enhance cellular resistance and can reduce fitness costs caused by resistance such as by decreased spore production and reduced spore germination [101,102,103]. In our study, we found that metabolism-related genes were enriched that focused on sugar metabolism, lipid metabolism, and amino acid metabolism, and several special genes were found, such as LCB1/2 and FRC1, verifying that cellular autophagy and related metabolism occur [104,105]. Therefore, it is hypothesized that the changes to related genes within RII strains compensate for fitness costs by increasing the expression of metabolism-related genes to promote cellular utilization of resources and energy production to maintain normal physiological function and survival of the cells.

In conclusion, significant differences in mitophagy and peroxisome expression were found between the RI and RII strains. Therefore, we hypothesized that there is an acting pathway between mitophagy, peroxisomes, and ROS, and that the difference in this pathway leads to the difference in resistance to QoIs between RI and RII strains. We suggest that the formation of the difference in resistance between RI and RII strains is related to the following: (i) ROS created by QoIs’ action on the mitochondrial electron transport chain are degraded to harmless substances by SOD, CAT, and GSH within peroxisomes, which reduces the cellular damage and improves resistance to QoIs. (ii) Mitophagy, induced by mitochondrial damage, is carried out with phagophores and lysosomes (Figure 11). Re-formation of mitochondria is promoted to maintain cellular homeostasis, thereby preserving mitochondrial physiological function and enhancing resistance to QoIs. And the lysosomes and endocytosis are involved in the above physiological processes of synthesis and metabolism. In contrast, mitophagy and ROS degradation may not have occurred in the RI strains, resulting in the accumulation of ROS, which caused irreversible damage to the cellular structure and reduced resistance to QoIs. This study reveals that mitophagy and peroxisomes are involved in the development of fungal resistance at the transcriptome level, providing new ideas and theoretical support for fungal resistance mechanisms.

## Figures and Tables

**Figure 1 microorganisms-11-02849-f001:**
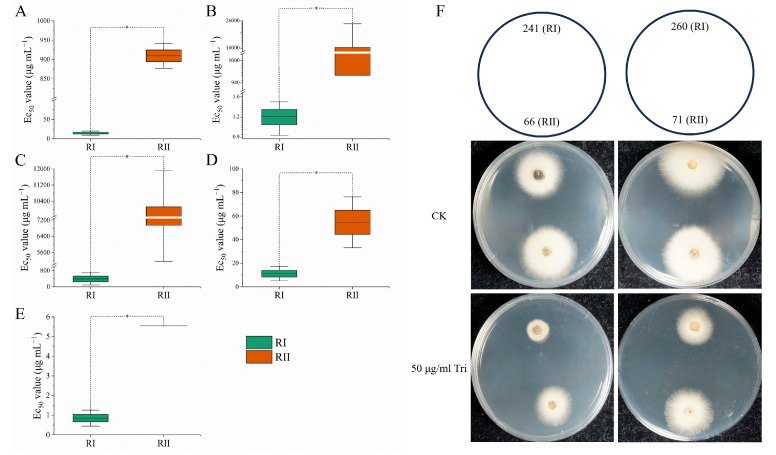
The sensitivity of RI and RII strains to QoIs. (**A**–**E**): EC${}_{50}$ for trifloxystrobin (**A**), kresoxim-methyl (**B**), cyazofamid (**C**), azoxystrobin (**D**), and pyraclostrobin (**E**). (**F**): The growth of RI and RII strains on pure PDA plates and on PDA plates with 50 μg/mL trifloxystrobin. * Significance level was *p*-value < 0.05 using Duncan’s test of one-way ANOVA.

**Figure 2 microorganisms-11-02849-f002:**
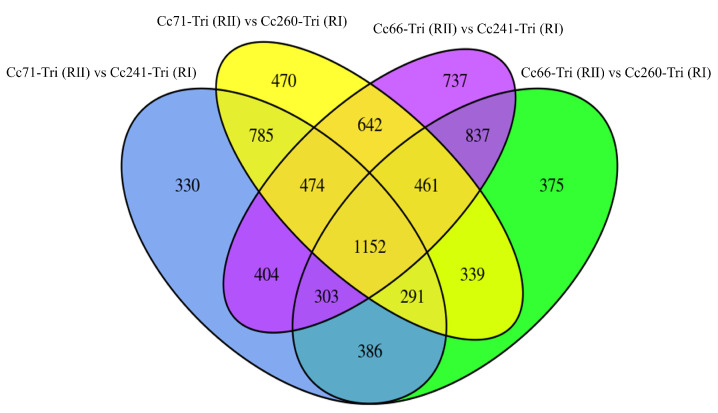
Venn graph of hepatopancreas transcriptome in Cc71-Tri (RII) vs. Cc241-Tri (RI), Cc71-Tri (RII) vs. Cc260-Tri (RI), Cc66-Tri (RII) vs. Cc241-Tri (RI), and Cc66-Tri (RII) vs. Cc260-Tri (RI). Venn graph of DEGs with different colors indicate four different groups. The numbers in the overlapping part represent the numbers of DEGs shared between groups, while the numbers in non-overlapping parts represent the numbers of DEGs unique to each group.

**Figure 3 microorganisms-11-02849-f003:**
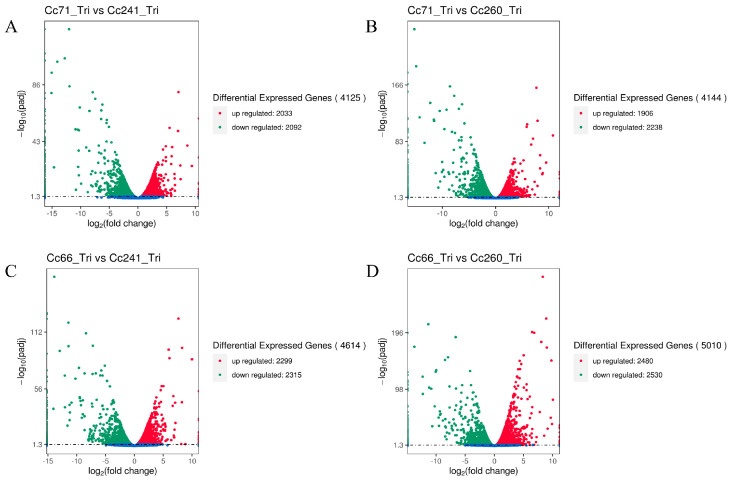
Volcano plot of DEGs in the comparison of Cc71-Tri (RII) vs. Cc241-Tri (RI) (**A**), Cc71-Tri (RII) vs. Cc260-Tri (RI) (**B**), Cc66-Tri (RII) vs. Cc241-Tri (RI) (**C**), and Cc66-Tri (RII) vs. Cc260-Tri (RI) (**D**). The up-regulated, down-regulated, and unchanged unigenes are dotted in red, green, and blue, respectively.

**Figure 4 microorganisms-11-02849-f004:**
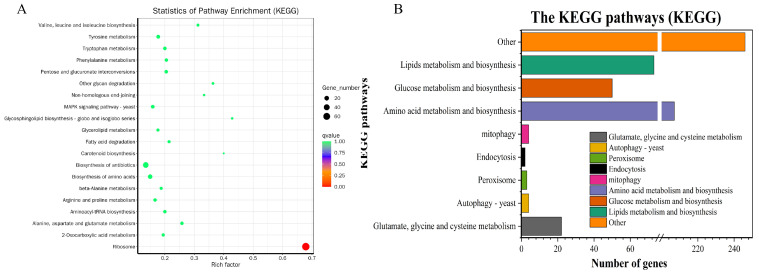
Distribution of KEGG pathway enrichment for RI and RII strains. (**A**): The top 20 enriched KEGG pathways; (**B**): enrichment of mitophagy, peroxisome, and endocytosis. Other: all enriched pathways other than those listed above. The y-axis represents the KEGG pathway name, and the x-axis indicates the number of enriched DEGs.

**Figure 5 microorganisms-11-02849-f005:**
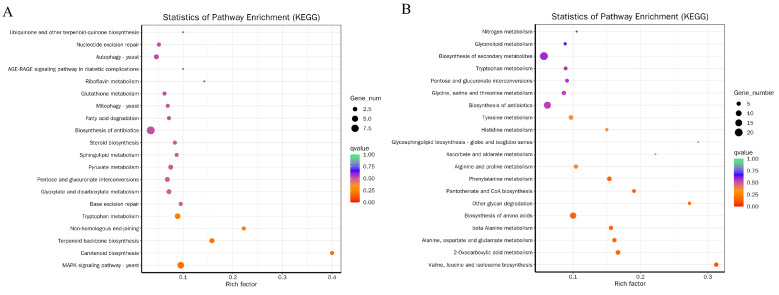
Distribution of up-regulated (**A**) and down-regulated (**B**) DEGs in the comparison between RI and RII strains. The y-axis represents the KEGG pathway name, and the x-axis indicates the enrichment factor.

**Figure 6 microorganisms-11-02849-f006:**
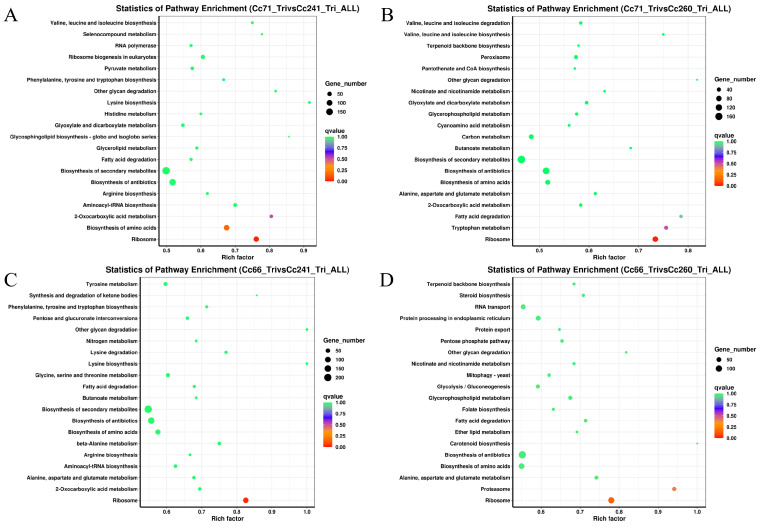
KEGG pathway enrichment. Distribution of DEGs for the comparison of Cc71-Tri (RII) vs. Cc241-Tri (RI) (**A**), Cc71-Tri (RII) vs. Cc260-Tri (RI) (**B**), Cc66-Tri (RII) vs. Cc241-Tri (RI) (**C**), and Cc66-Tri (RII) vs. Cc260-Tri (RI) (**D**). The y-axis represents the KEGG pathway name, and the x-axis indicates the enrichment factor.

**Figure 7 microorganisms-11-02849-f007:**
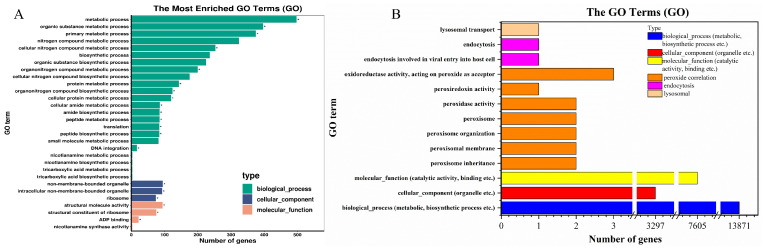
Gene ontology (GO) classifications of DEGs for RI and RII strains. (**A**): For the overall DEGs, GO enrichment classified them into three main categories: biosynthetic process, cellular component, or molecular function. (**B**): With reference to KEGG enrichment, DEGs were classified into one of six categories: peroxide correlation, endocytosis, lysosomal, biological process (BP, other metabolic, biosynthetic process, etc.), cellular component (CC, organelle, etc.), or molecular function (MF, other catalytic activity, binding, etc.). “*” indicates significant enrichment (*p*-value < 0.05).

**Figure 8 microorganisms-11-02849-f008:**
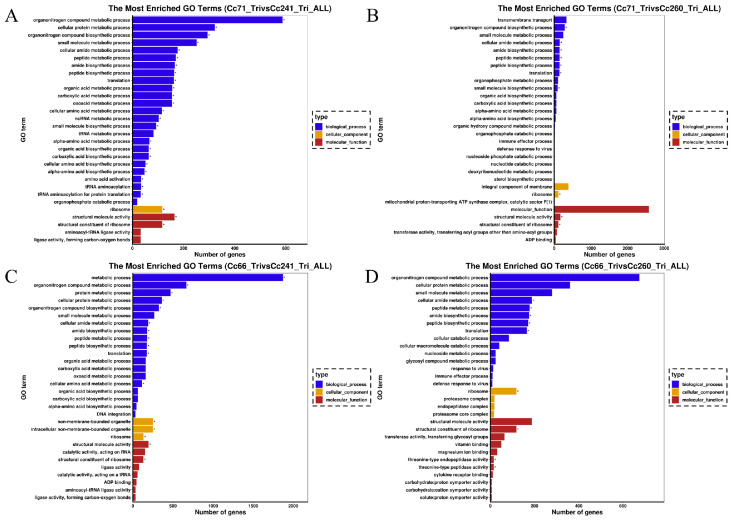
Geneontology (GO) classifications of DEGs for the comparisons of Cc71-Tri (RII) vs. Cc241-Tri (RI) (**A**), Cc71-Tri (RII) vs. Cc260-Tri (RI) (**B**), Cc66-Tri (RII) vs. Cc241-Tri (RI) (**C**), and Cc66-Tri (RII) vs. Cc260-Tri (RI) (**D**). For each comparison, GO enrichment classified DEGs into one of three categories: biological process, cellular component, or molecular function. “*” indicates significant enrichment (*p*-value < 0.05).

**Figure 9 microorganisms-11-02849-f009:**
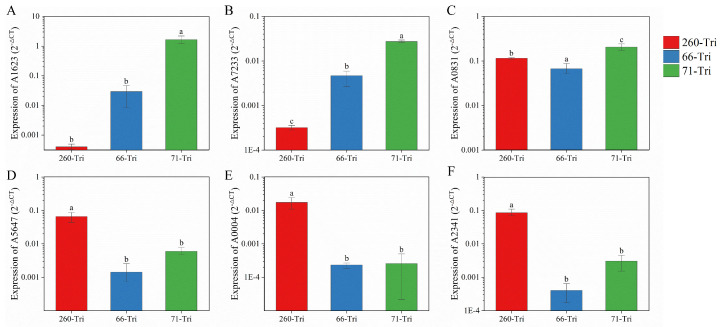
The validation of transcriptomic data with qPCR. The expression of the A1623 (**A**), A7233 (**B**), A0831 (**C**), A5647 (**D**), A0004 (**E**), and A2341 (**F**) gene in bacteria 260-Tri, 66-Tri, 71-Tri strains is indicated, respectively. The relative expression levels were calculated and analyzed by using the 2${}^{-\Delta Ct}$ method; a, b, and c indicate the significant differences (*p*-value < 0.05).

**Figure 10 microorganisms-11-02849-f010:**
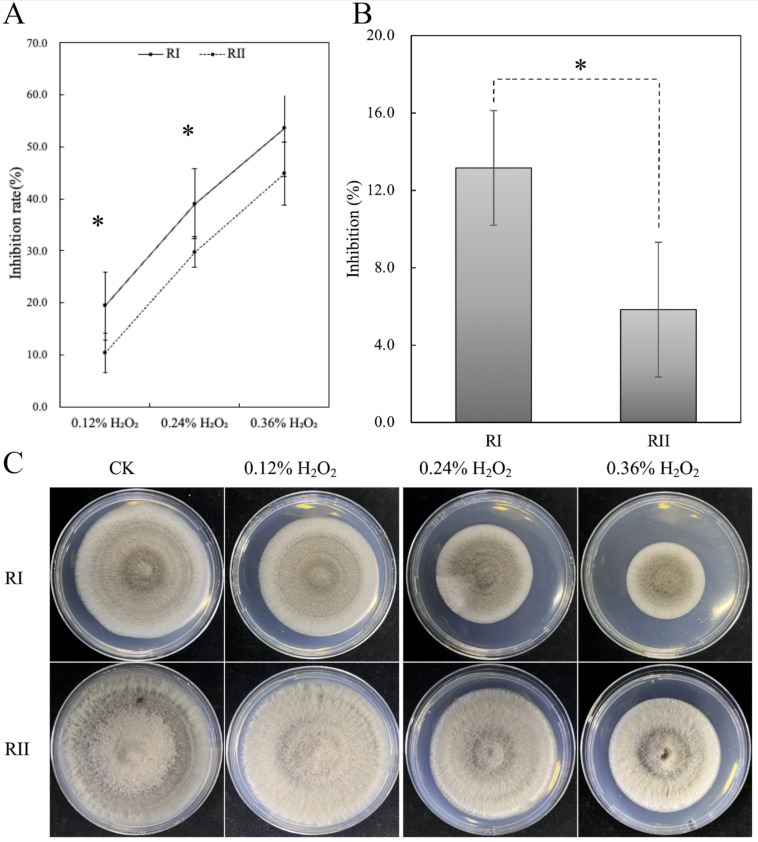
The growth of RI and RII strains under oxidative stress. (**A**) The inhibition rate of RI and RII strains at H${}_{2}$O${}_{2}$ concentrations of 0.12%, 0.24%, and 0.36%. (**B**) The inhibition rate of RI and RII strains with paraquat. (**C**) Growth of RI and RII type strains on plates with different concentrations of H${}_{2}$O${}_{2}$. *: Significance level was *p*-value < 0.05 using Duncan’s test of one-way ANOVA.

**Figure 11 microorganisms-11-02849-f011:**
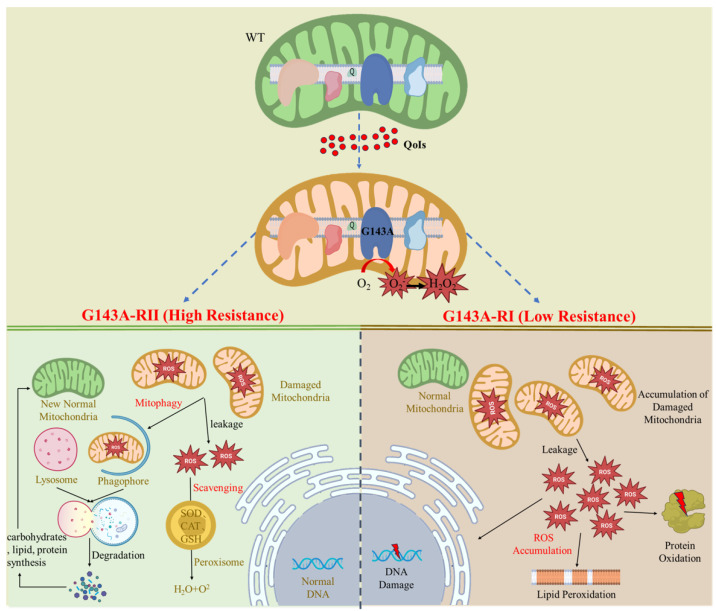
Suspected mechanisms for the differential resistance of RI and RII strains to QoIs. It is suggested that QoIs action caused a high production of ROS, and that scavenging mechanisms such as mitophagy and peroxisomes functioned in RII strains to prevent oxidative stress, whereas RI strains were less capable of resisting oxidative stress, resulting in different resistance to QoIs. The brown font represents organelles, and the black font represents physiological processes.

**Table 1 microorganisms-11-02849-t001:** Summary of transcriptome assembly.

Sample	Raw Reads	Clean Reads	Clean Bases	Q20 (%)	Q30 (%)	GC (%)
Cc260-Tri-1	52,805,726	52,025,868	7.8 G	97.91%	94.31%	57.21%
Cc260-Tri-2	50,946,058	50,132,632	7.52 G	97.94%	94.44%	57.36%
Cc260-Tri-3	51,392,366	50,619,174	7.59 G	98.11%	94.91%	57.24%
Cc241-Tri-1	54,353,210	53,409,354	8.01 G	97.80%	94.18%	56.82%
Cc241-Tri-2	56,572,142	55,709,416	8.36 G	97.94%	94.50%	56.93%
Cc241-Tri-3	67,258,428	66,177,436	9.93 G	97.93%	94.51%	56.73%
Cc66-Tri-1	58,396,696	57,283,728	8.59 G	97.88%	94.39%	56.96%
Cc66-Tri-2	63,712,892	62,669,002	9.4 G	97.69%	93.88%	57.04%
Cc66-Tri-3	54,820,830	53,534,274	8.03 G	97.95%	94.57%	57.04%
Cc71-Tri-1	61,482,532	60,734,460	9.11 G	97.89%	94.39%	56.64%
Cc71-Tri-2	58,387,148	57,468,612	8.62 G	97.98%	94.67%	56.80%
Cc71-Tri-3	53,542,268	52,693,836	7.9 G	97.99%	94.68%	56.89%

## Data Availability

The data presented in this study are available on request from the corresponding author.

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
