# Peer review of "Transcriptome Analysis Reveals the Involvement of Mitophagy and Peroxisome in the Resistance to QoIs in *Corynespora cassiicola"

_microorganisms, 2023, doi:10.3390/microorganisms11122849_

Round 1

Reviewer 1 Report

Comments and Suggestions for Authors

This research article investigated the resistance mechanism of two fungi mutants to fungicides (QoIs). The results showed that the selected mutants RI and RII showed a different level of resistance, even though they both had the same mutation. The resistance was shown to be due to enhanced mitophagy and peroxisomes scavenging ROS produced by the fungicide. This work allows to better understand the resistance mechanism of Corynespora casiicola to fungicides widely used in agriculture.

Minor comments:

Lines 85-87: Please add references: "Previous studies".

What do you mean by "However, the observed differences could not be solely attributed to fungal mechanisms? "

Figure 1 is not a graphical abstract since it does not give an overview of the research article. It could be used in discussion to explain the difference between RI and RII, but not in the introduction. Additionally, please add a title to the figure.

Line 108: what do you mean by "required"? I think it's "added"

Figure 4: "The up-regulated, down-regulated, and unchanged unigenes are dotted in red, and green, respectively" >> red, green, and blue (third color)

Figures in general are too small and thus hard to read.

Comments on the Quality of English Language

The manuscript is generally well-written. Please check some mistakes and sometimes unfinished sentences.

Example line 129: "Modification of the purified double-stranded cDNA"  unfinished sentence

Author Response

Thank you for your reviewers’ comments concerning our manuscript entitled “Transcriptome analysis reveals the involvement of Mitophagy and Peroxisome in the Resistance to QoIs in Corynespora cassiicola”. We appreciate and fully accept the proposed changes. They have been revised and highlighted in red in the document and the responses are highlighted in red below.

1. Point-by-point response to Comments and Suggestions for Authors

Comments 1: Lines 85-87: Please add references: "Previous studies".

Response 1: Thank you for the suggestion. The phenomenon was found in our lab’s research but not published. Therefore, I have changed this sentence in lines 86-87: “In our previous study, we have identified two phenotypes of C. cassiicola (RI and RII) with G143A mutations, which exhibited significantly varying resistance to QoIs”.

Comments 2: What do you mean by "However, the observed differences could not be solely attributed to fungal mechanisms?

Response 2: Thank you for your question and suggestion. This sentence was a poor expression on my part and was originally intended to express that common fungal resistance mechanisms could not explain the differences in resistance of the strains in the article. Therefore, I have revised the sentence in lines 87-90: “However, it was found to be impossible to explain the emergence of this phenomenon by common fungicide resistance mechanisms (target gene point mutation, alternative respiration, overexpression of transporters)”.

Comments 3: Figure 1 is not a graphical abstract since it does not give an overview of the research article. It could be used in discussion to explain the difference between RI and RII, but not in the introduction. Additionally, please add a title to the figure.

Response 3: Thank you for your suggestion. The original Figure 1 has been moved to the appropriate place in the discussion and the title and comments have been added. Currently listed in Figure 11.

Comments 4: Line 108: what do you mean by "required"? I think it's "added"

Response 4: Thank you for your suggestion. I have revised the word and checked for similar issues throughout the text.

Comments 5: Figure 4: "The up-regulated, down-regulated, and unchanged unigenes are dotted in red, and green, respectively" >> red, green, and blue (third color).

Response 5: Thank you for the suggestion, it was indeed an oversight on our part. I have made an addition to the sentence: “The up-regulated, down-regulated, and unchanged unigenes are dotted in red, green, and blue respectively”.

Comments 6: Figures in general are too small and thus hard to read.

Response 5: Thanks for the suggestion. Have enlarged all figures in the article.

3. Response to Comments on the Quality of English Language

Point 1: The manuscript is generally well-written. Please check some mistakes and sometimes unfinished sentences. Example line 129: "Modification of the purified double-stranded cDNA" unfinished sentence

Response 1: Thank you for your suggestion, we have proofread and revised the language throughout the text and have changed lines 137-138 to: “Then, double-strand cDNA was purified by using AMPure XP beads. Modification of the purified double-stranded cDNA”.

Reviewer 2 Report

Comments and Suggestions for Authors

Here are some overall comments and suggestions that could further improve this paper:

  • The introduction could be refined by more clearly distinguishing past work on medical drug resistance versus resistance in plant fungal pathogens specifically. This would sharpen the context.
  • Consider moving some supplemental results to the main text (certain pathway analysis charts, oxidative stress details) to better showcase the key data.
  • Discuss whether the expression changes observed may compensate for fitness costs of the target mutations. This could provide insight into selective advantages.
  • Elaborate on the proposed model - are there any validated inhibitors of these pathways that could be tested to further support the conclusions?
  • Carefully proofread the manuscript to fix minor typos, formatting issues, and any unclear wording.
  • Some sections of the results could be condensed by focusing on the most salient points and removing redundant descriptions.

-Acknowledge any limitations of solely analyzing RNA expression changes to infer protein activity and functional mechanisms.

Author Response

Thank you for your reviewers’ comments concerning our manuscript entitled “Transcriptome analysis reveals the Involvement of Mitophagy and Peroxisome in the Resistance to QoIs in Corynespora cassiicola”. We appreciate and fully accept the proposed changes. They have been revised and highlighted in red in the document and the responses are highlighted in red below.

1. Point-by-point response to Comments and Suggestions for Authors

Comments 1: The introduction could be refined by more clearly distinguishing past work on medical drug resistance versus resistance in plant fungal pathogens specifically. This would sharpen the context.

Response 1: Thank you for your advice. The presentation of the transition paragraph between medical drug resistance and fungicide resistance has been adjusted, focusing on lines 53-57: “In medicine, mechanisms such as target gene mutation and overexpression of transporters have also been reported. However, there is a common resistance mechanism in medicine that has not been reported more in fungicides. This extremely important mechanism is that cancer cells can regulate drug resistance by modulating physiological processes and organelles such as mitophagy and peroxisome, for which ROS is a major inducer”.

Comments 2: Consider moving some supplemental results to the main text (certain pathway analysis charts, oxidative stress details) to better showcase the key data

Response 2: Thank you for your advice. After consideration, we have decided to move the two supplemental figures to their corresponding places in the article, currently Figure 6 and Figure 8.

Comments 3: Discuss whether the expression changes observed may compensate for fitness costs of the target mutations. This could provide insight into selective advantages.

Response 3: Thank you for your suggestion. The discussion has been supplemented to target this point, focusing on lines 378-390: “Metabolism-related genes are widely involved in cellular energy metabolism, substance synthesis, catabolism, and transport, which can affect the cell’s ability to adapt to the external environment. It has been demonstrated that polysaccharide metabolites, lipid metabolites, and certain functional genes, such as the R gene, can enhance cellular resistance and can reduce fitness costs caused by resistance such as decreased spore production and reduced spore germination. In our study, we found that metabolism-related genes were enriched focusing on sugar metabolism, lipid metabolism, and amino acid metabolism, and several special genes were found, such as LCB1/2 and FRC1, verifying that cellular autophagy and related metabolism occur. Therefore, it is hypothesized that the changes of related genes within RII strains compensate fitness costs by increasing the expression of metabolism-related genes to promote cellular utilization of resources and energy production to maintain normal physiological function and survival of the cells”.

Comments 4: Elaborate on the proposed model - are there any validated inhibitors of these pathways that could be tested to further support the conclusions?

Response 4: Thank you for your question. In this article, oxidative stress capacity has been measured, validating the pathway to some extent. We are also currently conducting experiments on the production and activity of peroxidase and related proteins.

Comments 5: Carefully proofread the manuscript to fix minor typos, formatting issues, and any unclear wording.

Response 5: Thank you for your advice. We have proofread the text fully for formatting, italics, symbols, spelling, etc., e.g., lines 30 (“is a crucial fungicide” to “are crucial fungicides”), 83 (“Colorectal” to “colorectal”), 161 (“EF-1 alpha” to “EF-1 alpha”), and 235 (supplement of “RⅠ”).

Comments 6: Some sections of the results could be condensed by focusing on the most salient points and removing redundant descriptions.

Response 6: Thanks for the suggestion. The results section has been streamlined or supplemented, focusing on the GO and KEGG enrichment analysis.

For example, the redundant part " which were enriched in 547, 534, 456, and 451 DEGs, respectively " has been deleted.

The sentence " Similarly, two-by-two comparative KEGG analyses were performed for each strain in which peroxisomes, glutamate, glycine, cysteine metabolism, autophagy, and endocytosis were enriched. The highest enrichment factor was all in ribosomes and the highest number of enriched DEGs were in metabolic pathways " has been optimized to “Similarly, two-by-two comparative KEGG analyses were performed for each strain in which the highest enrichment factor was all in ribosomes and the highest number of enriched DEGs were in metabolic pathways”.

Comments 7: Acknowledge any limitations of solely analyzing RNA expression changes to infer protein activity and functional mechanisms.

Response 7: Thanks for the advice and the reminder. We very much agree with you. However, we consider that RNA is able to reflect protein changes to some extent and is informative about the results. Some linguistic adjustments and additions have been made with regard to that issue.

Reviewer 3 Report

Comments and Suggestions for Authors

Overall this is a good paper.  It is well written and addresses an important area.  A few points to consider are as follows:

- on line 17 and 18 I believe RI and RII strains need to be reversed.  This is obviously a big difference!

- on P. 3 list the modes of action of each fungicide

- on P. 5 list the sensitivity of each fungicide

-In Fig. 4 it says "up-regulated, down-regulated, and unchanged  genes are dotted in red and green, respectively".  This lists 3 groups and only 2 colors so meaning is not clear.

- There is no description of statistics used and how multiple tests were analyzed.  Also no indication of regression statistics used to calculate ec50 values (goodness of fit)   

Comments on the Quality of English Language

There were a few cases of in appropriate use of terminology.  For example P. 2 refers to "regulating physiological processes such as mitophagy and peroxisome".   Peroxisomes are organelles, not physiological processes.  On P. 3 the fungicides used are describes as "original drugs".  A better term would be "technical grade active ingredient"  which I assume is what was meant.  However, I am uncertain since no solvent was mentioned.  This is an important point.  If formulated product was used (not requiring a solvent), the specific formulation must be reported.

P. 5 "different fold"  would better be said by "4 times higher" as done in the following lines

Author Response

Thank you for your reviewers’ comments concerning our manuscript entitled “Transcriptome analysis reveals the involvement of Mitophagy and Peroxisome in the Resistance to QoIs in Corynespora cassiicola”. We appreciate and fully accept the proposed changes. They have been revised and highlighted in red in the document and the responses are highlighted in red below.

1. Point-by-point response to Comments and Suggestions for Authors

Comments 1: on line 17 and 18 I believe RI and RII strains need to be reversed.  This is obviously a big difference!

Response 1: Thank you for your advice. There may have been a misunderstanding due to an error in the resistance mechanism Figure. The RⅡ type strain is indeed more resistant, while the RⅠ strain is less resistant. We have now changed the Figure 11 and proofread the whole article for this issue.

Comments 2: on P. 3 list the modes of action of each fungicide

Response 2: Thank you for your suggestion. A brief addition to fungicide action has been made in lines 101-110 of the text: “To determine the susceptibility of RI and RII strains to QoIs, there were 4 fungicides were selected: trifloxystrobin (Tri), kresoxim-methyl (Kre), azoxystrobin (Azo), and pyraclostrobin (Pyr), which targeted to the Qo site of the cytochrome bc1 complex. Additionally, 7 other fungicides were selected to determine susceptibility: cyazofamid (Cya, targeted to the Qi site of the cytochrome bc1 complex), penthiopyrad (Pen, targeted to the ubiquinone binding site), terbinafine (Ter, Inhibits lanosterol 14α-demethylase), fludioxonil (Flu, involved in the high-osmolarity glycerol (HOG) stress response signal transduction pathway), tolnaftate (Tol, inhibits ergosterol production), difenoconazole (Dif, inhibits fungal lanosterol-14α-demethylase activity and blocks ergosterol biosynthesis), and carbendazim (Car, hindering microtubule assembly and disrupting spindle formation)”.

Comments 3: on P. 5 list the sensitivity of each fungicide.

Response 3: Thank you for your suggestion. Sensitivity data have been added in lines 183-197 of the text: “The average EC50 values of RII strains were 909.44, 8565.47, 54.69, and 5.55 μg/ml for 4 QoIs (Tri, Kre, Azo, and Pyr), while the average EC50 values of RI strains were 14.49, 1.21, 11.15, and 0.85 μg/ml for 4 QoIs, respectively. Among them, the largest difference in the resistance was observed in Kre, followed by Tri, with the average EC50 of RII strains being 7102.38 and 62.78 times higher than that of RI strains. And the resistance of RII strains to Azo and Pyr was 4.91 and 6.51 times higher than that of RI strains. However, there was no significant difference in resistance between RI and RII strains to the other 7 fungicides (Cya, Tol, Ter, Flu, Dif, Car, Pen), with the average EC50 values were 8529.00, 1.46, 0.37, 0.12, 2.01, 194.90, and 8.88 μg/ml for RII strains and 389.50, 23.69, 0.67, 0.12, 2.88, 341.72, and 0.80 μg/ml for RI strains. Note that due to the extreme resistance of Cya, the EC50 data are only for reference (formulaic extrapolation based on inhibition at low concentrations).

Comments 4: In Fig. 4 it says "up-regulated, down-regulated, and unchanged genes are dotted in red and green, respectively". This lists 3 groups and only 2 colors so meaning is not clear.

Response 4: Thanks for the suggestion, which was indeed an oversight on our part. I have made an addition to the sentence: “The up-regulated, down-regulated, and unchanged unigenes are dotted in red, green, and blue respectively”.

Comments 5: There is no description of statistics used and how multiple tests were analyzed.  Also no indication of regression statistics used to calculate ec50 values (goodness of fit)

Response 5: Thank you for your question and suggestion. EC50 values were calculated by probit regression using SPSS software. The significant analyses of variance were performed by the Duncan test for one-way ANOVA with a significance level was P<0.05.

This has been supplemented in the Materials and Methods section of the article, specifically in lines 124 and 167.

2. Response to Comments on the Quality of English Language

Point 1: There were a few cases of in appropriate use of terminology. For example P. 2 refers to "regulating physiological processes such as mitophagy and peroxisome". Peroxisomes are organelles, not physiological processes.

Response 1: Thank you for your suggestion. We have proofread and revised the language throughout the text and have changed lines 55-57 to: “This extremely important mechanism is that cancer cells can regulate drug resistance by modulating physiological processes and organelles such as mitophagy and peroxisome, for which ROS is a major inducer”.

Point 2: On P. 3 the fungicides used are describes as "original drugs". A better term would be "technical grade active ingredient" which I assume is what was meant. However, I am uncertain since no solvent was mentioned. This is an important point. If formulated product was used (not requiring a solvent), the specific formulation must be reported.

Response 2: Thank you for your suggestions. The content has now been changed and solvents added. It is clearly in lines 111-115: “Among them, Car was dissolved with 0.2M HCL and the other fungicides were dissolved with DMSO. The concentrations of active ingredients used were as follows: Kre and Cya (0, 0.1, 1, 10, 100, and 300 μg/mL), Azo and Pyr (0, 0.1, 1, 10, 80, and 240 μg/mL), Ter (0, 0.001, 0.01, 0.1, 1, and 5 μg/mL), Flu (0, 0.005, 0.01, 0.05, 0.1, and 0.5 μg/mL), Dif (0, 0.05, 0.1, 0.5, 3, and 9 μg/mL), Car (0, 10, 50, 200, and 400 μg/mL), Pen (0, 0.1, 0.5, 1, 3, and 10 μg/mL)”.

Point 3: P. 5 "different fold" would better be said by "4 times higher" as done in the following lines

Response 3: Thank you for your suggestion, we have proofread and revised the language throughout the text. The main change is in lines 188-189: “And the resistance of RII strains to Azo and Pyr was 4.91 and 6.51 times higher than that of RI strains”.